# Recurrence after Curative Resection for Intrahepatic Cholangiocarcinoma: How to Predict the Chance of Repeat Hepatectomy?

**DOI:** 10.3390/jcm10132820

**Published:** 2021-06-26

**Authors:** Serena Langella, Nadia Russolillo, Paolo Ossola, Andrea-Pierre Luzzi, Michele Casella, Roberto Lo Tesoriere, Alessandro Ferrero

**Affiliations:** General and Oncological Surgery, Mauriziano Hospital, Largo Turati 62, 10128 Turin, Italy; nrussolillo@mauriziano.it (N.R.); paolo.ossola@uniroma1.it (P.O.); andreapierre.luzzi@gmail.com (A.-P.L.); michelecasella2@gmail.com (M.C.); rlotesoriere@mauriziano.it (R.L.T.); aferrero@mauriziano.it (A.F.)

**Keywords:** intrahepatic cholangiocarcinoma, repeat resections, recurrences

## Abstract

(1) Background: Tumor recurrence after liver resection (LR) for intrahepatic cholangiocarcinoma (ICC) is common. Repeat liver resection (RLR) for recurrent ICC results in good survival outcomes in selected patients. The aim of this study was to investigate factors affecting the chance of resectability of recurrent ICC. (2) Methods: LR for ICC performed between January 2001 and December 2020 were retrospectively reviewed. Patients who had undergone first LR were considered for the study. Data on recurrences were analyzed. A logistic regression model was used for multivariable analysis of factors related to RLR rate. (3) Results: In total, 140 patients underwent LR for ICC. Major/extended hepatectomies were required in 105 (75%) cases. The 90-day mortality was 5.7%, Clavien–Dindo grade 3, 4 complications were 9.3%, N+ disease was observed in 32.5%, and the median OS was 38.3 months. Recurrence occurred in 91 patients (65%). The site of relapse was the liver in 53 patients (58.2%). RLR was performed in 21 (39.6%) patients. Factors that negatively affected RLR were time to recurrence ≤12 months (OR 7.4, 95% CI 1.68–33.16, *p* = 0.008) and major hepatectomy (OR 16.7, 95% CI 3.8–73.78, *p* < 0.001) at first treatment. Survival after recurrence was better in patients who underwent RLR as compared with not resected patients (31 vs. 13.2 months, *p* = 0.02). (4) Conclusions: Patients with ICC treated at first resection with major hepatectomy and those who recurred in ≤12 months had significantly lower probability to receive a second resection for recurrence.

## 1. Introduction

Recently, the incidence of intrahepatic cholangiocarcinoma (ICC) has been increasing worldwide, particularly in Western countries, reaching a peak of 0.85 per 100,000 [1,2]. Many patients with ICC present with advanced disease, thus, only 20–40% of patients with ICC are considered to be potentially resectable at diagnosis [3,4]. As the systemic chemotherapy response is low, surgery remains the gold standard in ICC treatment and the only curative chance. However, the five-year survival rate is still low (20–40%) [4,5] with a mean survival of 12–36 months after curative surgery, versus 2–9 months for the unresectable patients [6]. The low survival rate is mainly related to the disease recurrence after surgical resection that can be as high as 50–70% [7], being mainly localized in the liver [8], followed by distant lymph node metastases. Therefore, nowadays, ICC liver recurrence treatment is the challenge. Surgery for recurrent ICC seems to be a rational treatment option [9], albeit no well-defined guidelines are currently available. In recent years, several studies have demonstrated a survival benefit of repeat liver resection (RLR) for recurrent ICC [8,9,10,11,12,13,14] as compared with non-surgical treatment. A recent, multicentric German study [15] identified tumor biology and radicality of surgery at first resection, as strong determinants for resectability in the case of relapse. Nevertheless, data predicting the chance of RLR have been poorly investigated. The aim of this study is to assess factors affecting the reliability of RLR in the case of recurrent ICC.

## 2. Materials and Methods

We performed a retrospective analysis of data recorded in a prospectively maintained database for all consecutive patients with histologically confirmed ICC who had undergone liver resection (LR) with curative intent between January 2001 and December 2020. Patients with mixed hepatocellular cholangiocarcinoma were excluded. Demographic and clinical data were collected. The collection and registration of the original database were performed according to regulations and with the approval of the institutional review boards of our hospital, and written informed consent was obtained from all participants.

### 2.1. Patients Management

The initial workup of patients presenting with suspected ICC included liver function tests, carcinoembryonic antigen and CA 19.9, abdominal ultrasonography, and thoracoabdominal computer tomography (CT) scan. Abdominal magnetic resonance imaging (MRI) was performed in cases of doubt of biliary infiltration. Additional positron emission tomography (PET) was performed in cases of suspected extrahepatic metastases.

Patients were considered to be non-resectable in the case of extrahepatic metastases, distant lymph node metastases (e.g., paraortic), or peritoneal carcinomatosis. Multifocal disease was not an absolute contraindication to surgery.

PVE was planned when the future liver remnant volume estimated on the basis of the scheduled hepatectomy was <25% in patients with a normal liver, <30% in those with chronic liver disease [16] and <40% in those with cirrhotic liver [17]. LR was performed 4 weeks later if there had been sufficient hypertrophy.

Jaundice patients underwent preoperative biliary drainage (percutaneous transhepatic biliary drainage or endoscopic biliary drainage) in the case of planned major or extended hepatectomy requiring portal vein embolization (PVE), as well as signs of cholangitis and malnutrition (serum albumin <3 g/dL and/or unintentional weight loss of >10% of body weight over a 6-month period).

Laparoscopic exploration with intraoperative ultrasonography (LUS) to rule out peritoneal carcinomatosis, liver metastases, and vascular infiltration was performed routinely at the beginning of the operation [18] In patients scheduled for PVE, a staging laparoscopy with LUS was also carried out before PVE. The surgical technique of both open [19] and laparoscopic [20] hepatectomy were previously described. 

In the case of vascular infiltration, when major venous resection was required, the reconstruction was accomplished by peritoneal patch [21] or end-to-end anastomosis, as appropriated.

Adjuvant chemotherapy (capecitabine) was frequently given in the case of lymph node involvement. Patients were followed every 3 months for the first two year and biannually thereafter. A diagnosis of recurrence was established at radiological imaging. In particular, radiological follow-up was performed alternating abdominal ultrasonography and CT scan. MRI was also made in the case of diagnostic doubt at CT scan, while FGD-PET was performed in selected cases to confirm the suspicion of recurrence or in patients with increased levels of Ca 19.9.

### 2.2. Management of Recurrences

All cases of recurrence after hepatectomy were discussed by a multidisciplinary team. Upfront redo-surgery was considered for patient with one-site relapse, who presented with a disease-free interval longer than 12 months. Patients who recurred within 12 months after surgery were addressed to systemic chemotherapy. Those who did not progress at the restaging after chemotherapy were scheduled for surgery. Local treatments were considered for elderly and frail patients with localized recurrence. Patients with multi-metastatic or extended disease were addressed to systemic chemotherapy or best supportive care according to performance status. 

### 2.3. Definitions

Types of hepatectomy were classified according to the Brisbane 2000 terminology [22]. Major hepatectomy was defined as resection of three or more Couinaud segments. Extended hepatectomy was defined as resection of five or more Couinaud segments. Resection was considered complete (R0) when a tumor was resected with a surgical margin ≥1 mm. In the case of a surgical margin width <1 mm or tumor exposure at cutting edge, resection was considered to be incomplete (R1). In patients with multiple lesions, we counted on the smallest margin to define the R status. Operative mortality was defined as death before discharge from the hospital or within 90 days after surgery. Morbidity included any deviation from the normal postoperative course, and major morbidities were defined as any grade III or higher complication according to the classification scheme proposed by Dindo et al. [23]. Satellite nodules were defined as tumor nodules of ≤2 cm in diameter located ≤2 cm from the primary tumor. Multifocality was defined as more than one nodule in different Couinaud segments [24]. Tumor size was defined as the largest diameter of the tumor in the resected specimen; in the case of multiple tumors, the largest lesion was used as the index lesion. All specimens were reviewed according to the criteria of the eighth edition of the *AJCC Cancer Staging Manual* for TNM staging. 

### 2.4. Statistical Analysis

The entire statistical analysis was performed with IBM SPSS Italy (v.20.0). Normality was tested with the Kolmogorov–Smirnov test (*p* < 0.001). Categorical variables were compared using the chi-square test, Fisher’s exact test, or Pearson’s test as appropriate. Continuous variables were compared using the unpaired *t*-test or the Mann–Whitney U as appropriated. Continuous variables were presented as medians with range or interquartile range (IQR). Categorical variables were represented as number and percentage in brackets. Uni- and multivariate binary logistic regression analyses were performed to assess the predictive factors affecting the reliability of RLR. After the univariate analysis, a *p*-value ≤0.05 was considered to include variables in the multivariate analysis. The time to first recurrence was measured from the date of hepatic resection until the date of first recurrence. Disease-free survival was measured from the date of hepatic resection until the date of the first untreated recurrence. Overall survival was measured from the date of hepatic resection until the date of death or last follow-up. For patients who recurred, overall survival was also measured from the date of recurrence until the date of death or last follow-up. In particular, the follow-up date of living patients was censored.

The Kaplan–Meier method was used to estimate survival probabilities, which were compared using the log-rank test. All *p*-values were two sided, and *p* ≤ 0.05 was considered to be statistically significant.

## 3. Results

During the study period, 154 patients with ICC were scheduled for hepatectomy; 14 patients (9%) were considered to be unresectable at staging laparoscopy, nine patients because of peritoneal carcinomatosis and five patients due to LUS findings. Thus, 140 patients underwent first resection for ICC and were included in the study. 

Half of the patients (n = 71) were male with a median age of 66.5 years. The median BMI was 25 Kg/m^2^. In 32.1% of the cases, ICC occurred in patients with liver steatosis. In 16 patients, previous HBV or HCV infections were observed, while metabolic syndrome was present in 17 patients (12.1%). Hepatolithiasis or cirrhosis were associated in four patients (2.9%) and six patients (4.3%), respectively. Patient characteristics are detailed in Table 1. 

A major or extended hepatectomy was required in 75% of the patients, but the rate of major hepatectomies changed over the years decreasing from 80% in the first decade to 66% in the last ten years. Forty-eight patients (34.3%) needed associated resection. Resection of the main bile duct was performed in 33 cases (23.6%) and venous resections in 11 cases (7.9%). Overall, 83 patients experienced post-operative complications (59.3%). Among these, 13 cases (9.3%) were major complications. The 90-day mortality was 5.7%. The perioperative data are summarized in Table 2.

Gross-type was mainly mass forming (85.7%) with a median diameter of 60 mm (10–270 mm). Mixed-type was observed in 17 patients (12.1%), while the remnant 3 patients had an intrabiliary growth. In 93% of the cases, the resected tumor was >5 cm in size with associated satellitosis in 19.3% of the cases. Multifocal disease was present in 27 cases (19.3%), among these 24 (88.9%) were bifocal.

Lymphadenectomy was performed in 123 patients (87.9%), with a median of eight harvested lymph nodes. Among 123 patients who underwent lymphadenectomy, 40 patients (32.5%) were N+ at the pathological examination. The pathological results are summarized in Table 3. After surgery, adjuvant chemotherapy was administered in 48 patients (34.3%). All patients were treated with a capecitabine-based chemotherapy regimen.

### 3.1. Long-Term Results and Pattern of Recurrences

After a median follow-up of 88.7 months, the median overall survival was 38.3 months (95% CI 27.7–48.9). 

Overall survival at 3, 5, and 10 years after hepatectomy were 51.2%, 38.5%, and 21.5%, respectively (Figure 1). Ninety-one (65%) patients recurred after surgery. 

The median disease-free survival was 24.2 months (95% CI 17–31.4), while the time to first recurrence was 11.2 months (95% CI 8.9–13.6) (Figure 2). Fifty-three (37.8%) patients recurred within 12 months from hepatectomy. Early relapse (≤6 months) was observed in 15.7% of the cases. 

The liver was the only site of recurrence in 58.2% of the cases. In nineteen patients (20.9%), the site of relapse was extrahepatic. Lymph nodes were the most common (57.9%) sites of extrahepatic disease.

Intra- and extrahepatic recurrences were both observed in 19 patients (20.9%) (Table 4).

Among the patients who recurred in ≤12 months, the main site of relapse was the liver only (51.9%). Nevertheless, in this subgroup, both intra- and extrahepatic recurrences were observed in 32.7% of the cases, while extrahepatic only was observed in 15.4% of the cases. A different pattern of distribution was seen among 21 patients with early relapse. In this group, the more common pattern of presentation was both intra- and extrahepatic (42.8%) recurrences. Recurrences within the liver or extrahepatic only were registered in 33.4% and 23.8% of the cases, respectively. 

In patients with late (>12 months) recurrence, the site of relapse was the liver in 66.6% of the cases, extrahepatic in 28.2% of the cases, and both intra- and extrahepatic in 5.2% of the cases. The characteristics of recurred patients according to the time of relapse are detailed in Table 5. 

### 3.2. Management of Recurrences

Among 91 patients who recurred after hepatectomy, 39 patients (42.9%) received systemic chemotherapy-only treatment. Redo-surgery for recurrence was performed in 26 cases: 21 RLR for liver recurrence, three surgical procedures for localized lymph node disease, and two lung resections.

Among 53 patients who presented a relapse within 12 months from first hepatectomy, 28 patients (52.8%) had a liver-only recurrence. Of these, 18 patients (64.3%) were addressed to neoadjuvant chemotherapy, and finally six patients (33.3%) underwent hepatectomy after chemotherapy.

Overall, 65 patients were not resected. Among these, 19 patients (29.2%) were not scheduled for surgery because of concomitant intra- and extrahepatic relapse, 14 patients (21.5%) because of the site (i.e., bone, peritoneal, etc.) or number of extrahepatic recurrences, and 32 patients (49.3) because of multifocal liver-only relapse, of these 12 were potentially resectable at diagnosis but progressed after systemic chemotherapy. Locoregional treatment or radiotherapy were indicated in 10 cases, while 16 patients were placed in supportive care (Table 4).

Survival after recurrence was significantly better in patients who underwent RLR as compared with not resected patients 31 months (95% CI 10.2–51.7) vs. 13.2 months (95% CI 10.3–16.0), *p* = 0.02) (Figure 3).

In particular, 1-and 3-year OS were 84% and 37% among patients re-resected vs. 56.9% and 6% of patients who did not undergo RLR, respectively.

Patients who underwent surgery for extrahepatic recurrences had a median OS of 88.2 (95% CI 57.6–118.7), higher than not-resected patients with extrahepatic ony relpase 35.2 (9.8–60.6), *p* = 0.011

Interestingly, patients who underwent surgical treatment for any type of recurrence presented a 3-year OS of 72.9%, similar to those who had never developed a relapse 75.9%, *p* = 0.220. Similar results were also observed for RLR (3-year OS 76.4%, *p* = 0.156) as compared with patients who did not recur. Among 49 patients who did not recur after first resection, eight patients died during follow-up. In three cases, death was related to the progression of other primary malignant tumor, while, in the remnant cases, five patient deaths were non-cancer related (three cardiovascular accidents, one pulmonary disease, and one traumatic death)

### 3.3. Repeat Hepatectomy Details

Twenty-one patients underwent RLR for liver recurrence. One patient received a third hepatectomy for re-recurrence. They were 13 females (61.2%) with a median age of 69 years (range 23–85 years). In all cases, it was a recurrence of a mass-forming gross-type. The median time to recurrence was 21 months (range 4–84 months). 

Mainly, minor hepatectomies (20 of 21) were performed. Seven liver resections (33%) were accomplished by laparoscopic approach, among these, four patients had previously underwent open hepatectomy at first resection. Associated resections were required in three cases: one splenopancreatectomy, one resection of the main bile duct, and one venous resection of both right hepatic vein and inferior vena cava with reconstruction by peritoneal patch. One patient who did not received lymphadenectomy at first hepatectomy underwent lymphadenectomy during the RLR. 

Mortality was nil. Three patients experienced post-operative complications (14%), two were minor complications, in one case it was a biliary fistula requiring operative management by endoscopic biliary drainage.

### 3.4. Predictors of Repeat Hepatic Resection

The impact of 25 pre-intra and post-operative variables on the chance of resectability in the case of hepatic recurrence was studied. A univariate analysis showed that patients who underwent major hepatectomy (*p* < 0.001) or associated resection of the main bile duct at first treatment (0.052), with positive lymph nodes at first hepatectomy (*p* = 0.027), and a time to recurrence of ≤12 months (*p* < 0.001) significantly reduced the possibility of RLR. At multivariate analysis, factors that negatively affected the chance of RLR were a time to recurrence of ≤12 months (OR 7.4, 95% CI 1.68–33.16, *p* = 0.008) and major hepatectomy (OR 12.1, 95% CI 2.43–60.74, *p* = 0.002) at first treatment (Table 6).

As a result, among 50 patients with recurrence treated by major or extended hepatectomy at first procedure only, 6 patients (12%) were re-resected. Similarly, only 13.3% of the patients who recurred within one year from first hepatectomy had RLR.

## 4. Discussion

Long term surgical outcomes of ICC are negatively affected by the high rate of post-hepatectomy recurrence [25,26]. To improve long-term results, repeat resection in the case of recurrent ICC has been advocated [9]. The feasibility of redo-surgery after a first LR for ICC has been widely demonstrated [10,11,12,13,14,15,26,27,28]. Our data suggested that RLR may be, in some cases, technically demanding, requiring additional procedures such as vascular or biliary reconstruction. Nevertheless, when performed at tertiary centers, post-operative outcomes are acceptable and RLRs are safe. Si et al. [14] reported that only 2.8% of patients developed grade III/IV Clavien–Dindo complications after 72 RLR for recurrence, and 1 patient (1.4%) died during the perioperative period. In our experience, mortality after RLR was nil and major complications were less than 5%. The safety of RLR might be influenced by selecting patients in whom a repeat liver resection can be safely performed. Interestingly, we noticed that one-third of patients could benefit from a laparoscopic approach even when the first LR was previously performed by open surgery.

Several studies have demonstrated the efficacy of repeat surgical resection for the treatment of recurrent ICC [10,11,12,13,14,15,26,27,28]. The median survival after RLR ranges between 10 and 66 months. This heterogeneity could be related to different sites and number of treated metastases, timing of surgery, and association or not with systemic chemotherapy [9]. 

In the present study, patients who underwent RLR in the case of recurrence had a better prognosis as compared with not resected patients with a median survival of 31 months after RLR as compared with 13 months for those who were not resected. The timing of recurrence seems to be a key factor to better select patients who can benefit from redo-surgery. Patients who recurred within 6 months from LR had a low chance of re-resectability and presented worse result [29]. In the present series, up-front surgical treatment was offered only to patients who had a disease-free interval longer than one year. The use of preoperative chemotherapy also appears to be rational to improve oncologic results after RLR. In this series, patients who recurred within 12 months after the first LR were systematically addressed to systemic chemotherapy. In this way, chemotherapy may be able to select patients with favorable biology that can subsequently benefit from RLR. 

The site and number of relapses also had an impact on survival. Yoh et al. [30] reported a 3-year survival rate of 86.7% in 15 patients who underwent surgery mainly for lung recurrence. Similarly, in our series, five patients underwent surgery for isolated extrahepatic metastasis (lymph nodes or lung) with good outcomes. 

One of the largest multi-institutional series coming from Germany [15] included 113 RLR, resulting in a median survival of 36.8 months. The authors assessed factors associated with surgical resectability in the case of recurrent ICC and compared patients who underwent hepatectomy to those who had only repeated surgical exploration. They reported that the CA 19.9 level before primary resection, R status, and median time to recurrence significantly influenced resectability in the case of repeated exploration.

In this series, we focused on factors that could affect the chance of RLR in the case of hepatic recurrence. 

The multivariate analysis showed that patients who underwent major or extended hepatectomy at the first LR or recurred within 12 months from the first hepatectomy had a lower chance to be resected in the case of recurrence. 

As expected, the time to recurrence was confirmed to be a marker of unfavorable biology. In this light, we can speculate that a more aggressive relapse among patients with recurrence ≤12 months may explain the lower chance of RLR in this subgroup. 

Furthermore, after major hepatectomy, re-resection could not be performed in recurred patients with small remnant liver volume.

Because of the lack of specific symptoms, ICC frequently presented at diagnosis as advanced disease. For this reason, major or extended hepatectomies are frequently required to achieve complete resection [31]. Additionally, it is well known that long term outcomes are affected by R0 resection and margin width. Spolverato et al. [32] reported that patients who received a R1 resection had a higher risk of postoperative recurrence and shorter OS. In a study by Farges et al. [33], R1 resection was an independent predictor of poor survival in patients N0, while in N+ patients, survival was similar after R0 and R1 resections. Therefore, in our opinion it is reasonable to accept R1 in patient otherwise not resectable. 

It is well known that parenchymal-sparing surgery (PSS) is widely accepted for colorectal metastases to spare healthy liver and reduce major post-operative complications [34,35]. In recent years, the awareness that R1 vascular resection had a local disease control and survival equivalent to R0 resection [36], has pushed the PSS increasing the rate of resectability. This strategy cannot be fully applied to ICC because of different biology and pattern of presentation. Recently, Torzilli at al [37] compared results of resection for ICC with R1 resection (vascular and parenchymal) to R0. The R1 vascular and R1 parenchymal groups had similar overall and recurrence-free survival, which was significantly lower than the R0 group.

Nevertheless, it is important to keep in mind the possibility of re-resection during the first surgical procedure, even in patient with ICC, trying to avoid major hepatectomies when possible and still maintaining the oncological radicality. Because of these reasons, it is not surprising that the rate of major or extended hepatectomy in our series decreased over time. 

In summary, to define the surgical plan at first operation for ICC, it is important to consider tumor biology but also to evaluate the possibility of minor resections if oncological adequacy can be preserved. 

This study presents some limitations mainly related to its retrospective nature and the small sample size that was insufficient for more effective statistical analysis. A selection bias related to the better performance status of patients who underwent surgery may also have affected the results. For this reason, the influence of redo-resection on long-term outcomes cannot be exactly estimated

Moreover, we only focused on the results of resected patients. It would have been interesting to perform an intention to treat analysis to better understand the role of chemotherapy or other locoregional procedures. 

However, we confirm the key role of biology as a determinant of resectability for recurrent ICC and we firstly demonstrate that the extension of first hepatectomy for ICC may affect the possibility of re-resection for hepatic recurrence. 

## 5. Conclusions

In conclusion, RLR offers a survival benefit in selected patients with recurrent ICC. Patients with ICC treated at first resection with major hepatectomy, and those who recurred in ≤12 months from their first hepatectomy had a significantly lower probability of receiving a second resection for recurrence.

## Figures and Tables

**Figure 1 jcm-10-02820-f001:**
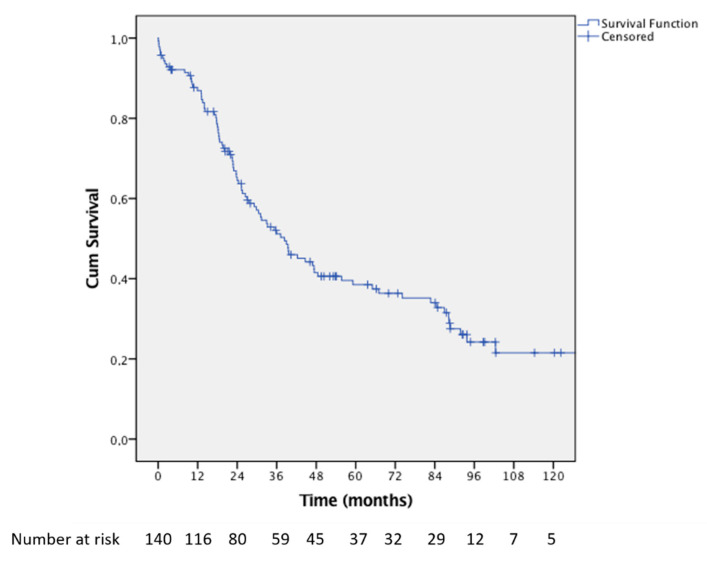
Kaplan–Meier curve showing the overall survival of the entire series from the date of first hepatectomy.

**Figure 2 jcm-10-02820-f002:**
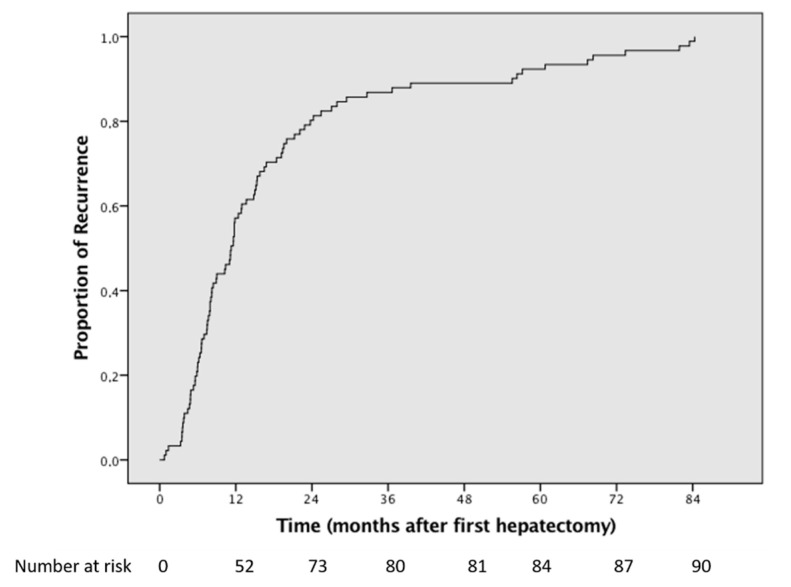
Kaplan–Meier curve showing the time to first recurrence after first hepatectomy.

**Figure 3 jcm-10-02820-f003:**
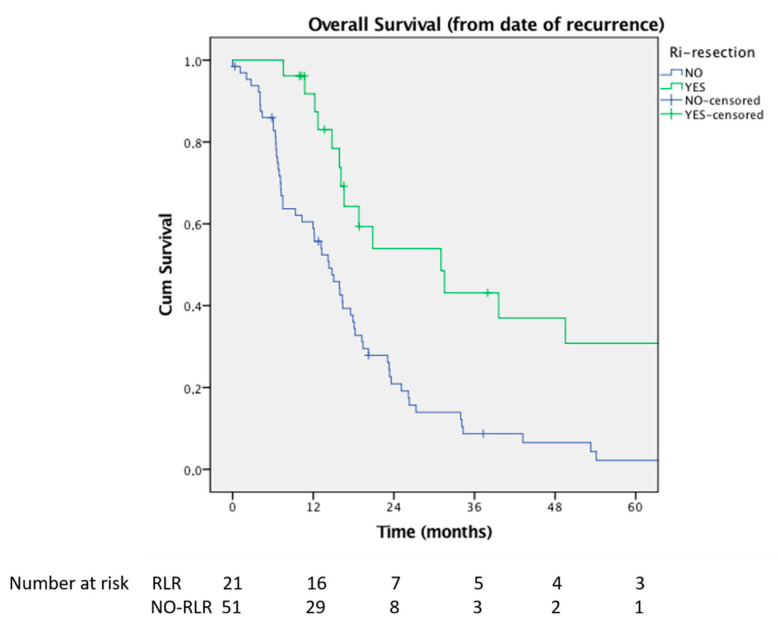
Comparison of Kaplan–Meier analyses of overall survival of the repeated resection group versus patients who had no resection for recurrence. Survival was calculated from the date of recurrence (*p* = 0.002).

**Table 1 jcm-10-02820-t001:** Patient characteristics.

Characteristics	*N* = 140
Age, years, median, (range)	66.5 (22–85)
Male, *n* (%)	71 (50.7)
BMI, kgm^2^, median, (range)	25 (17.9–41.4)
ASA 3-4, *n* (%)	65 (46)
Preoperative biliary drain, *n* (%)	13 (9.3)
Ca 19.9, U/mL, median (IQR)	28.7 (6.9–192.6)
Metabolic Syndrome, *n* (%)	17 (12.1)
HCV, *n* (%)	9 (6.4)
HBV, *n* (%)	12 (8.6)
Lithiasis, *n* (%)	4 (2.9)
Cirrhosis, *n* (%)	6 (4.3)
Steatosis, *n* (%)	45 (32.1)
NASH, *n* (%)	3 (2.1)

BMI, body mass index; ASA, America Society of Anesthesiology; NASH, non-alcoholic steatohepatitis.

**Table 2 jcm-10-02820-t002:** Perioperative Data.

Characteristics	*N* = 140
**Type of hepatectomy**	
Major or Extended hepatectomy, *n* (%)	105 (75)
Anatomic, *n* (%)	126 (90)
Anatomic + wedge, *n* (%)	7 (5)
Wedge, *n* (%)	7 (5)
**Associated Resection**	48 (34.3)
Biliary resection, *n* (%)	33 (23.6)
Vascular Resection, *n* (%)	11 (7.9)
Lymphadenectomy, *n* (%)	123 (87.9)
Laparoscopic Liver Resection, *n* (%)	14 (10)
**Postoperative complications**	83 (59.3)
Major complications, *n* (%)	13 (9.3)
Liver failure, *n* (%)	17 (12.1)
Biliary fistula, *n* (%)	30 (21.4)
Reintervention, *n* (%)	11 (7.9)
Mortality, *n* (%)	8 (5.7)
Hospital stay, days, median (range)	12.5 (2–114)

**Table 3 jcm-10-02820-t003:** Pathological results.

Characteristics	*N* = 140
Tumor diameter, mm, median (range)	60 (10–270)
Multifocal, *n* (%)	27 (19.3)
Grade of differentiation	
G1, *n* (%)	3 (2.1)
G2, *n* (%)	55 (39.3)
G3, *n* (%)	82 (58.6)
**AJCC 8th TNM**	
T1A, *n* (%)	9 (6.4)
T1B, *n* (%)	10 (7.1)
T2, *n* (%)	107 (76.4)
T3, *n* (%)	12 (8.6)
T4, *n* (%)	2 (1.4)
Perineural invasion, *n* (%)	52 (37.1)
Macrovascular invasion, *n* (%)	77 (55)
Microvascular invasion, *n* (%)	120 (85.7)
N+ *, *n* (%)	40 (32.5)
Harvested lymph nodes, median (IQR)	8 (4–13)
Surgical width, mm, median (IQR)	4 (1–9.5)
R1, *n* (%)	25 (17.9)

* Percentage calculated on 123 patients who underwent lymphadenectomy.

**Table 4 jcm-10-02820-t004:** Pattern and management of 91 recurrences among 140 patients who underwent first hepatectomy for ICC.

Site of Recurrences	*N* = 91
Liver only, *n* (%)	53 (58.2)
Extrahepatic, *n* (%)	19 (20.9)
Both, *n* (%)	19 (20.9)
**Management of Recurrences**	
Surgery, *n* (%)	26 (28.6)
Chemotherapy only, *n* (%)	39 (42.9)
Radiotherapy only, *n* (%)	5 (5.5)
Locoregional treatments, *n* (%)	5 (5.5)
BSC, *n* (%)	16 (17.6)

BSC, best supportive care.

**Table 5 jcm-10-02820-t005:** Characteristics of 91 patients with recurrences according to the time of relapse.

Characteristics	Very Early Recurrences (≤6 Months)	Early Recurrences (>6 Months, ≤12 Months)	Late Recurrences (>12 Month)	*p*
Male, *n* (%)	10 (47.6)	12 (38.7)	18 (46.2)	0.764
Age, years, median (range)	61 (29–76)	66 (43–83)	69 (22–85)	0.093
Multifocal	8 (38.1)	8 (25.8)	8 (15.4)	0.142
Preop. Ca 19.9, median (IQR)	190 (30.3–1602)	24.9 (9.6–74.1)	20 (4.1–209.7)	0.018 *
Size, mm, median (range)	80 (10–145)	70 (30–140)	50 (10–160)	0.024 *
T3-4, *n* (%)	2 (9.5)	4 (12.9)	6 (15.4)	0.813
G3, *n* (%)	15 (71.4)	17 (54.8)	24 (61.5)	0.483
N+, *n* (%)	10 (55.6)	13 (41.9	6 (19.4)	0.028 *
Microvascular infiltration, *n* (%)	18 (85.7)	29 (93.5)	35 (89.7)	0.646
Perineural infiltration, *n* (%)	12 (57.1)	13 (41.9)	9 (23.1)	0.027 *
Satellitosis, *n* (%)	4 (19%)	9 (29)	3 (7.7)	0.065
Margin width, mm, median (range)	5 (0–25)	5 (0–20)	4 (0–24)	0.181

* Very late vs. late.

**Table 6 jcm-10-02820-t006:** Uni- and multivariate analyses of factor determinants of resectability for ICC hepatic recurrences.

	Univariate Analysis		Multivariate Analysis		
	No RLR (*n* = 51)	RLR (*n* = 21)	*p*	OR (95% CI)	*p*
Age (years), median ± SD	66 ± 11.3	66.5 ± 14.3	0.616	-	
Male sex, *n* (%)	23 (45.1)	8 (38.1)	0.697	-	
Single tumor, *n* (%)	29 (56.9)	14 (66.7)	0.441	-	
Ca 19.9 preoperative	52 ± 8300	21.9 ± 1500	0.453		
Tumor diameter >50 mm, *n* (%)	36 (70.6)	13 (66.7)	0.743		
Major or ext. hepatectomy, *n* (%)	44 (86.3)	6 (28.6)	<0.001	12.1 (2.43–60.74)	0.002
Anatomical resection, *n* (%)	45 (88.2)	19 (90.5)	1.000	-	
Laparoscopic resection, *n* (%)	3 (5.9)	4 (19)	0.183	-	
Associated resection, *n* (%)	18 (35.3)	5 (23.8)	0.342		
Type of associated resection					
Biliary, *n* (%)	14 (27.5)	1 (4.8)	0.052	ns	
Vascular, *n* (%)	5 (9.8)	2 (9.5)	1.000	-	
Postoperative complications, *n* (%)	30 (58.8)	12 (57.1)	0.895	-	
Positive lymph node, *n* (%)	21 (46.7)	3 (16.7)	0.027	ns	
T stage (TNM 8th)				-	
-T1a, *n* (%)	0	0		-	
-T1b, *n* (%)	6 (11.8)	0	0.171	-	
-T2, *n* (%)	40 (78.4)	15 (71.4)	0.143	-	
-T3, *n* (%)	5 (9.8)	5 (23.8)	0.435	-	
-T4, *n* (%)	0	1 (4.8)	0.292	-	
Satellitosis, *n* (%)	12 (23.5)	1 (5)	0.093	-	
Surgical margin <1 mm, *n* (%)	9 (17.6)	4 (19)	1.000	-	
G3, *n* (%)	30 (58.8)	11 (52.4)	0.616	-	
Microscopic vascular infiltration, *n* (%)	46 (90.2)	20 (95.2)	0.482	-	
Major vascular infiltration, *n* (%)	28 (54.9)	13 (61.9)	0.585	-	
Adjuvant chemotherapy, *n* (%)	22 (43.1)	6 (28.6)	0.249	-	
Recurrence ≤12 months, *n* (%)	39 (76.5)	6 (28.6)	<0.001	7.4 (1.68–33.16)	0.008

## Data Availability

For access to data please contact the corresponding author (slangella@mauriziano.it).

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
