# Peer review of "Recurrence after Curative Resection for Intrahepatic Cholangiocarcinoma: How to Predict the Chance of Repeat Hepatectomy?"

_jcm, 2021, doi:10.3390/jcm10132820_

Round 1
Reviewer 1 Report
The manuscript by Langella et al. investigates the chance for repeat hepatectomy in case of recurrence in patients who underwent liver resection for intrahepatic cholangiocarcinoma. A cohort of 140 patients were included with 91 patients suffering from recurrence. repeat hepatectomy was performed in 21 patients and factors affecting the possibility to re-resect were time to recurrence less than 12 months and previous major hepatectomy.
In general, the manuscript is concise and interesting.
In addition, some other points should be addressed and are listed below:
Major points:
- Line 171: What is the difference between satellitosis and multifocal growth? This should be defined somewhere. Is there any data from literature that there is a difference between satellitosis and multifocality?
- The authors describe that in patients with early relapse the main site of recurrence is the liver (line 194). What about the late relapse group? I suggest to include a table that describes the characteristics of patients with very early relapse (< 6 months), early relapse (< than 12 months) and late (> than 12 months).
- In line 237/238 the authors state that the 3 years OS of patients who did not undergo re-resection is 0. However, the corresponding graph (fig. 3) does not show this result. Can you explain this?
- In line 345/346 the authors state that 5 patients with resection of isolated extrahepatic recurrence had good outcomes. Is there any data supporting this statement?
Minor points:
- Line 17 (abstract): “RLR was performed in 21 (39.6%).” I suggest to use the term “21 patients” instead of just 21.
- Line 20 (abstract): do these number really compare patients after repeat liver resection with untreated patients? 13 months without treatment is rather long. Are these maybe patients who underwent chemotherapy etc.?
Reviewer 2 Report
I have read with great interest this manuscript discussing the predictors for applicability of repeat hepatectomy in resected ICC patients. Although this paper is well-written, I have some major and minor issues to be concerned.
Major
- Page 2, line 85. Which regimen of adjuvant chemo Tx was used?
- Page 2, line 86-87. How often was the radiological test performed for checking the recurrence? And, which imaging modalities was used? These are very important points because the duration between first surgery and recurrence is the major issue in this study. Please explain.
- Page 3, line 90-93. Re-operation was applied for only patients who had recurrence at more than 12 months after first surgery. It is an inevitable result that recurrence within 12 month is a negative predictor for re-operation in multivariate analysis. What is authors' opinion?
- Page 4, line 176. What kind of adjuvant chemo Tx was used?
- Page 5, line 203-206. Among 92 patients who had recurrence, only 26 patients underwent re-operation, even though 53 patients had liver-only recurrence. Authors should describe the reasons why 66 (92 minus 26) patients could not undergo re-operation.
- Page 9, line 239-242. Authors noted patients who underwent re-operation for recurrence had a similar survival period to those who did not have recurrence. If so, what were the causes of death of patients without recurrence? Please mention about it.
- Page 9, line 266-273, and page 11, Table 5. Contents in the text are different from those of table 5. Time to recurrence and major hepatectomy are the significant negative factors for re-operation in the text. However, major hepatectomy and positive LN are the significant factors in table 5. Which is correct? Please revise its. Moreover, why is major hepatectomy a negative predictive factor for re-operation? Small remnant liver volume could not allow additional hepatectomy? Please explain in relation to the reason for incapability of re-operation in many recurrent patients.
- Page 10, line 295-297. Authors described that patients who underwent re-operation had a better prognosis than those who did not undergo re-operation. Because this study has a retrospective fashion, this comparison can be influenced by selection bias. Patients who underwent re-operation were considered to have a better general status than those who could not undergo re-operation, therefore, prognosis of former patients justifiably are better than those of latter patients. Contribution of re-surgery to prognosis cannot be estimated exactly. Authors should note this point.
Minor
- Page 1, line 25-31. This is an only template. Please remove it.
- Page 3, line 118. Median is usually used with range or interquartile range, not SD. SD should be used with mean.
- Page 3, line 122-125. I cannot understand the difference between "time to first recurrence" and "disease-free survival". Please explain more precisely.
- Page 3, line 125-127. In overall survival, last follow-up date of living patients should be classified as censored.
- Page 4, Table 1, 7th row. CA19-9 (median±SD) level is described as 28.7±7641. This is undoubtfully strange. Please re-calculate in median with (interquartile) range or mean with SD.
- Page 6, Table 3, 18th-19th rows. Median should be used with (interquartile) range, not SD.
- Page 7-8, Fig. 1, 2, 3. Kaplan-Meier curve should be expressed with number at risk.
Reviewer 3 Report
Intrahepatic cholangiocarcinoma (ICC) is the second most common primary liver cancer and surgical resection acts as a potential and available cure for patients with ICC. Liver resection should be individualized based on disease stage, liver function, and patient’s performance status.
In this study, the authors tracked and analyzed multiple parameters in the patients with repeat liver resection (RLR) and found that survival after recurrence was better in patients who underwent RLR compared to untreated patients. This study suggested that RLR is involved in extended the survival time in selected patients with ICC.
Some issues need to be addressed.
- In this study, tumor characteristics were showed in the pathological results of Table 3. Did the authors test the tumor size, number of tumors, American Joint Committee on Cancer eight edition N status and M status? Does tumor stage affect the survival time for patients with RLR?
- The authors found that survival time increased in the patients with recurrence ICC compares with non-treated patients. Besides increased survival, did the authors assess the life and health qualities of patients with RLR?
Round 2
Reviewer 2 Report
I have checked the revised version of manuscript.
The authors adequately corrected the manuscript according to the reviewer's comments in almost issues.
However, I have one point to be concerned.
In minor point 4, "median with IQR of CA19-9 was 28.7±185.73" is wrong.
IQR should be expressed as 〇-〇, not ±. ± is used for SD.
And the bottom quartile of CA19-9 should be positive number.
Please consult with statistical experts.
Author Response
Please see the attachmen
